# Endothelin and the Cardiovascular System: The Long Journey and Where We Are Going

**DOI:** 10.3390/biology11050759

**Published:** 2022-05-16

**Authors:** Andreas Haryono, Risa Ramadhiani, Gusty Rizky Teguh Ryanto, Noriaki Emoto

**Affiliations:** 1Division of Cardiovascular Medicine, Department of Internal Medicine, Kobe University Graduate School of Medicine, Kobe 650-0017, Japan; andreasharyono89@gmail.com; 2Laboratory of Clinical Pharmaceutical Science, Kobe Pharmaceutical University, Kobe 658-8558, Japan; helloimrr@gmail.com (R.R.); waniqmatun@gmail.com (G.R.T.R.)

**Keywords:** endothelin, endothelin receptor antagonist, pulmonary hypertension, heart failure, coronary artery disease, hypertension

## Abstract

**Simple Summary:**

In this review, we describe the basic functions of endothelin and related molecules, including their receptors and enzymes. Furthermore, we discuss the important role of endothelin in several cardiovascular diseases, the relevant clinical evidence for targeting the endothelin pathway, and the scope of endothelin-targeting treatments in the future. We highlight the present uses of endothelin receptor antagonists and the advancements in the development of future treatment options, thereby providing an overview of endothelin research over the years and its future scope.

**Abstract:**

Endothelin was first discovered more than 30 years ago as a potent vasoconstrictor. In subsequent years, three isoforms, two canonical receptors, and two converting enzymes were identified, and their basic functions were elucidated by numerous preclinical and clinical studies. Over the years, the endothelin system has been found to be critical in the pathogenesis of several cardiovascular diseases, including hypertension, pulmonary arterial hypertension, heart failure, and coronary artery disease. In this review, we summarize the current knowledge on endothelin and its role in cardiovascular diseases. Furthermore, we discuss how endothelin-targeting therapies, such as endothelin receptor antagonists, have been employed to treat cardiovascular diseases with varying degrees of success. Lastly, we provide a glimpse of what could be in store for endothelin-targeting treatment options for cardiovascular diseases in the future.

## 1. Introduction

The existence of a vasoconstrictor secreted by endothelial cells was first reported by several researchers as early as 1981 [1,2,3,4]. This culminates in 1988, where Yanagisawa et al. identified the vasoconstrictor peptide endothelin (ET; now known as endothelin-1 or ET-1) [5]. Endothelin-1 showed potent and long-lasting vasoconstrictor effects on arteries that were never observed with another compound at the time. Not long after the discovery of ET-1, two other isoforms have been discovered, each with distinct functions. These isoforms are known as endothelin-2 [6,7] (ET-2) and endothelin-3 [8] (ET-3). Two G-protein-coupled receptors, endothelin type A (ET_A_) [9,10] and endothelin type B (ET_B_) receptors [11,12], which can be activated when bound with endothelin peptides, were also identified. ET-1 and ET-2 are more potent than ET-3 in activating ET_A_, while all three isoforms are equipotent in activating ET_B_ [13]. Subsequently, two endothelin-converting enzymes (ECEs) that cleaved the endothelin precursor, “big endothelin”, into active peptides were identified (ECE-1 [14,15] and ECE-2 [16]). Since then, researchers have been working to uncover the role of the endothelin system in both health and disease. In this review, we discuss the basic knowledge of endothelin and its role in cardiovascular disease. Evidence of endothelin involvement in pathological conditions, both in preclinical and clinical studies, will be presented, and strategies to target this pathway as a therapeutic option in the past, present, and future will be discussed.

## 2. The Endothelin System

### 2.1. Biosynthesis of Endothelin

ET-1 belongs to the most abundantly synthesized endothelin peptide family. Mature ET-1 is a 21-amino-acid peptide with two cysteine bridges at the N-terminus and a free hydrophobic C-terminus. The crystal structure of ET-1 was solved recently using X-ray diffraction data collected in 1992 [17,18]. Endothelins have structures similar to snake venom toxins (safarotoxins), whose envenomation causes strong coronary artery constriction [19,20]. Endothelin receptor antagonists have been suggested as antivenoms [21]. Mature ET-1 peptide is synthetized by many types of cells, mainly vascular endothelial and smooth muscle cells, while macrophages, fibroblasts, podocytes, and brain neurons also express it [2,13]. Meanwhile, ET-2 peptide is synthetized mainly by intestinal epithelial cells, while it is also transiently expressed in the lung and ovarian follicles [7,22,23]. Finally, the ET-3 peptide is synthetized by melanocytes, intestinal cells, brain neurons, and other cells [2,24,25]. Endothelin peptide synthesis is activated in response to many factors such as hyperglycemia, hypercholesterolemia, aging, estrogen deficiency, hypoxia, shear stress, microRNAs, and angiotensin II [22,23,24,25].

Endothelin biosynthesis involves three steps, as illustrated in Figure 1. Endothelins are initially secreted as precursor 212 amino acid polypeptides, named preproETs. A signal peptidase cleaves the 17-amino acid signal to generate proETs, which are subsequently cleaved at the C and N terminals by furin enzymes to generate big ETs [25,26]. Finally, endothelin-converting enzymes (ECEs) cleave big ETs to produce mature ETs with 21 amino acids [14]. Because big ETs are biologically inactive, this maturation process is their key activity. Interestingly, in mice lacking both ECE-1 and ECE-2, mature endothelin peptide levels were reduced by one-third [27]. Other enzymes such as chymases are involved in the maturation of big ETs [28,29]. The deletion of chymases reduces mature endothelin levels [30,31], whereas overexpression increases it [32,33].

### 2.2. Endothelin Receptor

To activate its signaling pathways, the endothelin peptides bind to two subtypes of endothelin receptor, the ET_A_ receptor [9,10] and the ET_B_ receptor [11,12], which belong to the seven G-protein-coupled transmembrane-spanning domain receptors (GPCRs). Both ET-1 and ET-2 showed equal potency for the ET_A_ receptor binding, whereas ET-3 showed 100-fold lower affinity for the ET_A_ receptor. In contrast, ET-1, ET-2, and ET-3 showed similar potency to ET_B_ receptors [13,34]. ET_A_ receptor expression was relatively higher in the vascular smooth muscle, whereas ET_B_ receptor expression was higher in endothelial cells. Thus, ET_A_ and ET_B_ receptors are ubiquitously expressed in all organs that receive the blood supply. The ET_A_ receptor was expressed at the highest level in the lungs and heart, with lower expression in the brain, while the brain and periphery of the lung, such as capillaries, are rich in ET_B_ receptors [35].

ET_A_ receptor stimulation induced potent and prolonged vasoconstriction, inflammation, and cell proliferation, whereas ET_B_ receptor stimulation generally showed the opposite effects (see Figure 2) [34,36]. As such, the ET_B_ receptor can be considered an ET_A_ receptor endogenous antagonist. The ET_B_ receptor also functions in the clearance of ET-1 from circulation (see Figure 2) [37,38,39]. The crystal structure of the ET_B_ receptor and its interaction with ligands have been recently determined [40,41,42,43,44]. These findings shed light on the interaction between the ET_B_ receptor and its ligand as well as the underlying G-protein mechanism.

### 2.3. Endothelin Agonists and Antagonists

Numerous peptide and non-peptide compounds that act on endothelin receptors with varying degrees of potency and specificity have been discovered. Some of these compounds act as agonists and antagonists. Several compounds can act selectively, while others are non-selective on endothelin receptors [13]. Over the last two decades, the development of agonists and antagonists for endothelin receptors, ET_A_ and ET_B_, has been extensively studied. BQ123 and FR139317 were the first ET_A_-selective peptide antagonists to be identified. Parallelly, the ET_B_ agonists (BQ3020 and IRL1620) and the first selective antagonist peptide ET_B_ (BQ788) were identified. Within five years of the discovery of ET-1, a bioavailable non-peptide antagonist drug of the endothelin system was developed.

ET-1, ET-2, and ET-3 are agonists of the ET_A_ and ET_B_ receptors. However, because ET-3 has a lower affinity for the ET_A_ receptor, it is more likely to activate the ET_B_ receptor [45]. To date, no ET_A_ receptor agonists, either peptides or non-peptides, have been identified. It is generally accepted that the effects of ET_A_ activation in pathophysiological conditions are deleterious; therefore, there is no beneficial evidence for activating the ET-1/ET_A_ pathway [46]; however, several ET_B_ receptor agonists have been discovered to date. Sarafotoxin 6c, which has been used in experimental studies in humans, has notably high selectivity for rat ET_B_ receptors, but less so for human ET_B_ receptors [47,48]. IRL1620 [49] and BQ3020 [50] are the most widely used selective ET_B_ receptors. IRL1620 is used in experiments involving cerebral blood flow as a neuroprotective agent [51,52,53] and in cancers [54,55,56]. BQ3020 has been used in ET_B_ receptor characterization and labeling studies [57,58,59] and as a selective PET agent in vivo [60]. However, there is currently no evidence that agonist agents of endothelin have been initiated in cardiology.

On the contrary, endothelin receptor antagonists (ERAs) have been identified and utilized for several years. ERAs are classified as selective towards one receptor subtype or dual antagonists that block both ET_A_ and ET_B_ receptors. There is no agreement regarding the classification of these antagonists; however, Davenport and Maguire suggested that selective compounds should have more than 100-fold selectivity towards either ET_A_ or ET_B_ receptors, while those that display less selectivity than that are defined as balanced antagonists [61]. The clinical evidence of ERA use in cardiovascular diseases will be discussed in a later section.

Bosentan is the first antagonist of both ET_A_ and ET_B_ receptors and was approved by the U.S. Food and Drug Administration in 2001 for pulmonary arterial hypertension (PAH) [62,63]. The diverse side-effects of bosentan include headache, nasal congestion, flushing, fluid edema, elevated levels of liver enzymes, and anemia, which resemble those of ET_A_-selective antagonists. Bosentan-related elevation of liver enzymes is dose-dependent and typically asymptomatic [13,64,65,66]. Macitentan is a non-selective endothelin receptor antagonist that was approved for clinical use in PAH in 2013. It was designed by modifying the structure of bosentan to improve its efficacy and tolerability, resulting in reduced side effects, such as lower liver toxicity and lower extremity fluid retention, compared to bosentan [61]. Procitentan is a potent dual ET receptor antagonist derived from macitentan. Aprocitentan is currently under investigation for treatment-resistant hypertension, which will be discussed in detail later [67,68].

The most widely used ET_A_ receptor selective antagonist is BQ123 [69] which has been used in both in vivo and in vitro studies. Other peptide-based selective ET_A_ receptor antagonists used in experiments were FR139317 [70] and TAK-044 [71]. Ambrisentan [72] and sitaxentan [73] have been used in clinical trials to treat PAH patients. Ambrisentan was the second approved antagonist introduced in clinical settings for PAH treatment in 2007. However, in 2010, sixatentan was withdrawn owing to cases of idiosyncratic hepatitis resulting in acute liver failure and death [74]. Atrasentan [75], another highly selective ET_A_ receptor antagonist, has been successfully used in the treatment of diabetic nephropathy [76].

Selective ET_B_ receptor antagonists are less developed compared to other types of endothelin receptor antagonists, attributed to the potential danger of blocking ET-1 clearance and vasodilatation effects [13]. In the pre-clinical setting, the most extensively used ET_B_ antagonist is peptide BQ788 [77]. The last novel derivative from ERA is the relatively novel agent, sparcentan. Sparcentan is the first orally active antagonist with ET_A_ receptor and angiotensin II type 1 (AT_1_) receptor inhibitory activities in a single compound. It was developed by merging the elements present in the irbesartan AT_1_ receptor antagonists with elements in biphenylsulfonamide ET_A_ receptors. Currently, sparcentan has been investigated in several clinical trials related to kidney diseases [46,78].

### 2.4. Genetic Mutations in Endothelin System

Genetic mutations in endothelins, endothelin converting enzymes, and endothelin receptors have been shown to be involved in or risk factors for many diseases. For instance, mutations in endothelin 1 gene are associated with pediatric pulmonary hypertension [79], recessive auriculocondylar syndrome (ACS), and dominant isolated question-mark ears (QME) [80]. The rs9349379 SNP of the PHACTR1 locus (6p24), which is associated with coronary artery disease (CAD), migraine headache, cervical artery dissection, fibromuscular dysplasia, and systemic arterial hypertension [81], is a regulator of endothelin-1 expression [82].

Mutation in the ET_A_ receptor peptide-binding site alters its subtype selectivity, which affects its interaction with ligands [83]. Mutations in the ET_A_ receptor cause mandibulofacial dysostosis with alopecia [84]. The genetic variant of EDNRA, rs6841581, is significantly associated with an increased risk of intracranial aneurysm in East Asian populations [85,86,87]. The ET_A_ receptor (*ENDRA*-231 A/G) gene polymorphism is associated with migraine [88,89].

Mutations in endothelin-3 commonly affect the enteric nervous system and the melanocytes. As endothelin-3 exerts its function by interacting with the ET_B_ receptor, a similar phenomenon occurred in ET_B_ receptor mutations. Several mutations in endothelin-3 are associated with a combined Waardenburg type 2 and Hirschsprung phenotype (Shah-Waardenburg syndrome) [90,91,92]. Mutations in the ET_B_ receptor are also associated with Hirschsprung and Waardenburg syndromes [93,94,95,96]. Hypermethylation and downregulation of the ET_B_ receptor expression are associated with reduced patient survival and poor prognosis in several types of malignancies [97,98,99,100].

Not limited to the peptides or receptors only, mutations in the converting enzymes have also been linked to pathological conditions. The R742C mutation in the ECE-1 gene results in a patient with skip lesion Hirschsprung disease, cardiac defects, and autonomic dysfunction [101]. Another variation in ECE-1 is linked to essential hypertension [102].

### 2.5. Phenotype of Genetic Endothelin Modification in Mice

A whole-body ET-1 knockout mouse was developed by deleting exon 2 of the ET-1 gene [103]. Homozygous deletion (ET-1^−/−^) is lethal in neonates. Caesarian delivered mice on day 18.5, postcoital, all with major craniofacial and cardiac anomalies [103,104]. ET-1^−/−^ mice also have lower neonatal weight, poor thyroid and thymus development, and lesser cardiac sympathetic innervation [105,106]. Heterozygous deletion of ET-1 (ET-1^+/−^) resulted in different phenotypes in which the mice appeared normal, fertile, and with reduced ET-1 concentration in the lung and plasma. However, mice exhibit elevated blood pressure [103]. In the overexpression mouse model (ET-1^+^), the mice exhibited normal ET-1 in the blood, but increased ET-1 expression in the brain, lungs, and kidneys [107]. These mice exhibited chronic inflammation in the lungs [107]. Kidney phenotypes were more severe, exhibiting increased renal cyst formation, renal interstitial fibrosis, glomerulosclerosis, and age-dependent salt-sensitive hypertension [107,108,109,110].

Global ET-2 deletion in mice resulted in severe growth retardation, juvenile lethality, internal starvation, hypothermia, and abnormal lung histology. These findings revealed that ET-2 is important for postnatal growth and survival of mice by regulating energy homeostasis and maintaining lung function [111]. Global *ET-2* overexpression in Sprague-Dawley rats, called TGR(hET-2)37, results in male rats having significantly lower body weight accompanied by kidney interstitial and glomerular sclerosis. Female rats exhibit glomerulosclerosis [112,113].

ET-3 heterozygous mice (ET-3^+/−^) were phenotypically normal. However, global homozygous knockout mice (ET-3^−/−^) died early postnatally, with an average age of 21 days after birth. The mice also presented with aganglionic megacolon and coat color spotting. This result showed that ET-3 is required for the proper development of enteric neurons derived from the vagal neural crest and epidermal melanocytes derived from the trunk neural crest [114]. Piebaldism (absence of melanocytes in the skin) or lethal spotted (*ls*) phenotypes arose spontaneously in mouse colonies. These *ls/ls* mice also presented with megacolon. The *ET-3* transgene under the control of human dopamine-*β*-hydroxylase (D*β*H) introduced into *ls/ls* mice reduced piebaldism and megacolon in these mice. This evidence shows that the *ls/ls* mouse phenotype is a result of ET-3 deficiency [115].

ET_A_^−/−^ mice die shortly after birth due to severe craniofacial deformities and neural crest-derived structural abnormalities [116,117]. ET_B_^+/−^ mice appeared normal and were able to produce offspring. However, ET_B_^−/−^ mice were born healthy but became sick and died within 4 weeks, and showed similar abnormalities as ET-3^−/−^ mice, including megacolon and coat color changes [118]. *ECE-1* deletion resulted in mortality between embryonic day 12.5 (E12.5) and 30 min after birth. ECE-1^−/−^ mice showed cardiac and craniofacial anomalies identical to those in ET-1 and ET_A_ receptor-deficient mice [119]. On the other hand, ECE-2^−/−^ mice survive, appear healthy, fertile, and have the same lifespan as wild-type littermates. The simultaneous deletion of ECE-1 and ECE-2 with ECE-1^−/−^/ECE-2^−/−^ miceshowed broader and more severe cardiac abnormalities than ECE-1^−/−^ mice [27].

## 3. Endothelin in Cardiovascular Diseases

### 3.1. Pulmonary Hypertension

#### 3.1.1. Relations between Endothelin and PAH

Pulmonary hypertension (PH) was among the first conditions in which the clinical application of endothelin-targeting agents was tested. PH underwent a change in definition after the World Symposium on Pulmonary Hypertension 2018, where the threshold of the mean pulmonary artery pressure (mPAP) diagnostic criteria decreased from 25 mmHg to 20 mmHg [120]. PH is divided by the WHO into five groups based on etiology (pulmonary arterial hypertension/PAH, PH due to left heart disease, PH due to chronic lung disease or hypoxia, chronic thromboembolic PH/CTEPH, and PH due to other etiologies) [121]. However, a common thread linking the groups, although in varying degrees and locations, is the pulmonary vascular remodeling that causes an increase in pressure. This remodeling process primarily involves the dysfunction of the endothelial cells (EC) and smooth muscle cell (SMC) layers of the vessel, while the contributions of the adventitial layer of the vasculature and other surrounding cells are also noteworthy [122]. These dysfunctions include, but are not limited to, inappropriate vascular tone control, aberrant EC and SMC apoptosis, changes in proliferation capacity of all three vessel layers, and endothelial-to-mesenchymal transition [122,123].

Endothelin is a potent vasoconstrictor expressed in various vascular beds. For example, ET-1 is abundantly expressed in the lung. Due to this, ET is a prime candidate molecule to be involved in PH [34]. Various animal models of pulmonary hypertension have shown that an increase in both the cellular expression and circulating level of ET-1 could be found in chronic hypoxia (3 weeks of 10% O_2_), SU5416-hypoxia, and monocrotaline (MCT) models of PH, among others [124,125,126]. Further studies confirmed the mechanism of action by which endothelin could affect the vascular remodeling and dysfunction [127,128]. In addition to the well-known imbalance of the nitric oxide (NO) and prostacyclin (PGI_2_) vasodilation pathways due to the overactivation of ET, other important pathways are also affected by the binding of ET-1 to its receptors, ET_A_ and ET_B_ [129].

The expression pattern of ET_A_ and ET_B_ receptors in the lung vasculature varies according to the cells, where endothelial cells mainly express ET_B_, whereas smooth muscle cells and fibroblasts also express ET_A_ in addition to ET_B_ [130,131]. Accordingly, ET-1 affects various processes in these cells, ultimately causing vascular remodeling when overactivated. As mentioned, impaired balance of vasodilator (NO and PGI) and vasoconstrictors (e.g., thromboxane A_2_/TXA_2_) due to ET-1 is a major problem in the vasculature, while in smooth muscle cells, aberrant proliferation caused by activation of the PI3K, PLC, and MAPK pathways, in addition to being the effector site of the vasoconstrictive effects, can be observed [2,130]. Not limited to those effects, ET-1 overabundance has also been linked to increased EC apoptosis and decreased SMC apoptosis, the induction of a glycolytic switch in the EC, and the promotion of reactive oxygen species production, among others [132,133,134,135]. With a diverse array of pathways capable of being altered by this family of peptides, endothelin becomes vital to tackling the challenge of treating PH.

#### 3.1.2. Clinical Applications of Endothelin and ERAs in PH

With strong evidence of the involvement of endothelin, particularly ET-1, as demonstrated by several preclinical studies, the next important step was to check whether ET played an equally important role in patients. For this purpose, several studies were conducted, with results indicating an increase in ET-1 expression levels in the vascular endothelial cells of PAH patients and in the circulating levels of ET-1 in the blood [136,137]. Furthermore, it was found that the increased presence of ET-1 was not limited to PAH. Reports of ET-1 overabundance can also be found in PH due to left heart disease, PH due to lung disease, and CTEPH [138,139,140]. This underlines the importance of endothelin regardless of the etiological cause. Consequently, ET-1 has also been explored as a biomarker and especially as a prognostic tool. In patients with PAH, blood ET-1 levels have been shown to have prognostic value in predicting hospitalization and mortality [141].

The most important question regarding endothelin is whether targeting this pathway can translate into a beneficial treatment option. To this end, the blockade of the endothelin receptors ET_A_ and ET_B_ through the use of ERA is being tested in clinical trials. Among the groups of PH, PAH is currently the only condition in which the use of ERA is approved [142]. The introduction of ERA helped improve what was previously a bleak prognosis for PAH patients and improved its mortality and morbidity rates. The dual ET_A_/ET_B_ receptor antagonist bosentan was first approved as a treatment for PAH in the groundbreaking BREATHE-1 trial published in 2002 [63]. In this trial, 213 patients with primary or connective tissue disease-associated PAH were randomly assigned to either placebo or two different bosentan treatment regimens (125 mg twice daily or 250 mg twice daily) for a minimum of 12 weeks. Here, those treated with bosentan showed promising clinical improvements in the 6 min walk test distance, Borg dyspnea index, WHO functional class, and time to clinical worsening with tolerable levels of adverse effects. This study served as a major turning point in the clinical use of ERA, and further studies have confirmed its efficacy and improved upon the original BREATHE-1 trial. Bosentan, as the first dual ERA approved for clinical use, has also been studied in the PAH of various etiologies. For example, bosentan treatment in PAH due to HIV is beneficial, both in the short and long term, where both hemodynamic and clinical improvements can be seen [143,144]. Portopulmonary hypertension is another condition in which bosentan has been found to have similar clinical and hemodynamic benefits [145,146]. Lastly, in the case of PAH due to congenital heart disease (CHD), bosentan has been reported to be effective in patients with Eisenmenger syndrome in the BREATHE-5 trial [147,148].

Other ERAs with differing affinities to the two ET receptors, such as macitentan or ambrisentan, have also gained approval for use in patients with PAH in the last decade. Notably, the SERAPHIN trial analyzing macitentan usage in PAH patients revealed the benefits of this treatment [149]. In this trial, the investigators analyzed the efficacy of macitentan at two different dosages in comparison to placebo (3 mg or 10 mg), and found a decrease in the primary end-point event (death, lung transplantation, prostanoid treatment, atrial septostomy, or worsening PAH) occurrence rate [149]. Furthermore, macitentan was also recently reported to benefit right ventricular function and structure, in addition to improving hemodynamics in the REPAIR study [150].

In the case of ambrisentan, the ARIES set of clinical trials examined whether treatment with ambrisentan (5 mg or 10 mg doses in ARIES-1 and 2.5 mg or 5 mg doses in ARIES-2) compared to placebo could have beneficial effects [151]. The results showed that ambrisentan could effectively improve the clinical worsening of PAH, WHO functional class, Borg dyspnea index, and B-type natriuretic peptide levels. Notably, ambrisentan treatment did not exponentially increase liver enzyme levels by more than three-fold [151]. It is noteworthy that in both macitentan and ambrisentan, as is the case with bosentan, although there was significant hemodynamic and clinical improvement in comparison to placebo after treatment, the change was not significant enough to stop searching for ways to further improve PAH treatment.

The AMBITION trial answered the question of whether ERA could have a beneficial effect when combined with other PAH treatments [152]. In the AMBITION trial, ambrisentan was administered in combination with the PDE-V inhibitor tadalafil, and it was found that combined therapy with ambrisentan and tadalafil successfully reduced the rate of clinical worsening (death, hospitalization, worsening of PAH, disease progression, unsatisfactory response to treatment), while improving the NT-proBNP and 6 min walk test distance [152]. The TRITON trial attempted to determine whether the upfront triple combination therapy of ERA (macitentan), PDE-V-inhibitor (tadalafil), and prostacyclin receptor agonist (selexipag) is more beneficial than dual therapy (macitentan and tadalafil). Although no difference in pulmonary vascular resistance reduction was found between upfront double and triple therapy, an exploratory analysis showed a potential reduction in disease progression, albeit with a small sample size [153]. Obviously, ambrisentan and/or macitentan have also been analyzed for PAH due to various etiological causes, such as HIV, CHD, or hepatopulmonary hypertension, with varying degrees of success or lack thereof, in the case of the MAESTRO study on Eisenmenger syndrome [154,155,156].

Unfortunately, not all ERAs are suitable for the treatment of PAH. For instance, the trial for sitaxentan was terminated prematurely owing to the high incidence of liver dysfunction in the treatment arm [34,131]. Trials for newer ERAs have also hit a roadblock with slow recruitment, which was mentioned as the reason for the tezosentan trial in PAH conditions being terminated (NCT01077297). In summary, ERA has become an essential part of PAH treatment in the last decade and contributes to the improvement of patient prognosis.

Unfortunately, clinical trials for other forms of PH have not shown similar effectiveness for the use of ERA. Notably, ERAs failed to show sufficient evidence of efficacy in Group 3 PH due to lung disease or hypoxia, where several clinical trials did not achieve satisfactory results [157]. Notably, a single-center trial from the University Hospital Basel, Switzerland, which examined the use of bosentan in severe COPD, failed to show the benefits of additional ERA in both lung and cardiac functions [157]. Several trials analyzing the efficacy of ERAs in PH associated with idiopathic pulmonary fibrosis (IPF) also did not show a positive effect, and one study even observed that the administration of ambrisentan worsened the clinical condition of IPF patients [158,159]. This result is discouraging, especially considering the fact that ET-1 has been found to be a driver of the pro-fibrotic phenotype found in patients with IPF, both in preclinical and translational studies. However, it is noteworthy that the loss of a different endothelin isoform, ET-2, in the lung could worsen the IPF phenotype in a preclinical study [160]. As such, it might be plausible that different strategies are needed to target the endothelin pathway in this particular group of PH.

Another trial examining ERA use, i.e., bosentan in PH due to left heart disease, also failed to improve lung hemodynamics and RV remodeling [161]. Tezosentan similarly did not improve RV function in patients with a history of PH undergoing cardiac surgery [162]. Furthermore, several studies have shown the occurrence of liver injury due to ERA and fluid retention, which further complicates the use of ERA in this particular condition, such as those found in the MELODY-1 study [163]. These two adverse effects can also become major problems in the application of ERA in other cardiovascular conditions, such as hypertension and heart failure, which will also be discussed in another section.

Chronic thromboembolic pulmonary hypertension or CTEPH represents another condition in which ERA could potentially be used. However, to date, ERA has not been accepted as a treatment option for this condition. Even so, evidence of ERA utility in CTEPH has recently started to come to light. In 2008, the BENEFIT study analyzing the use of bosentan concluded that bosentan had a positive effect on hemodynamics, while no effect could be seen in exercise capacity [164]. Importantly, MERIT-1 reported that macitentan has significant clinical and hemodynamic benefits for patients with inoperable CTEPH [165]. Furthermore, recently, a recent multicenter study of macitentan use in CTEPH was terminated due to reasons unrelated to patient safety (the sponsor decided to discontinue the study). However, from the reported data, it appears that macitentan is a promising treatment option for CTEPH, although further studies are required to confirm this [166]. It is notable that a study found that ET_A_ expression was markedly increased in the thromboembolic lesions of CTEPH patients who underwent pulmonary endarterectomy (PEA) [167]. Focusing on blocking this receptor rather than using the traditional dual ET_A_/ET_B_ receptor blocker might be an interesting solution to treat CTEPH using ERA.

### 3.2. Systemic Arterial Hypertension

#### 3.2.1. ET in Basic Molecular Mechanism of Systemic Arterial Hypertension

Because of the nature of endothelin, which was originally found in the endothelium and acts as a vasoconstrictor in a delicate balance with other vasoactive peptides, ET is a highly interesting molecule for analysis in systemic arterial hypertension conditions [34]. Molecular mechanisms linking endothelin and systemic arterial hypertension have been discovered over the years. Various animal models of systemic arterial hypertension have shown that ET-1 levels increase during systemic arterial hypertension, while molecularly, as mentioned in the previous section, ET-1 has been shown to affect various pathways related to vascular tone control, such as the renin–angiotensin–aldosterone system (RAAS), nitric oxide (NO), prostacyclin, TXA_2_, cyclic guanosine monophosphate (cGMP)/cyclic adenosine monophosphate (cAMP), and adrenomedullin receptor activity modifying protein (RAMP) pathways [34,131]. Conversely, the loss of ET-1, specifically in vascular endothelial cells, could adversely affect vascular tone maintenance and cause systemic hypotension [168]. Several of the aforementioned pathways are not only known simply as modulators of vascular tone, but they are also known to be involved in endothelial dysfunction and arterial stiffness, i.e., two processes caused by the imbalance of said modulators. Indeed, it is now thought that endothelin has a wide range of effects beyond simple vasoconstriction. ET-1 is known to play a role in arterial stiffening. For instance, ET-1 is found to regulate pulse wave velocity and contribute to widening pulse pressure, while ET-1 has already been known to be closely related to NO production regulation, a major player in arterial stiffness [169,170]. Another study related ET-1 levels with IL-6, a known marker for arterial stiffness [171]. Lastly, ET-1 is also known to contribute to aging, i.e., another factor that contributes to arterial stiffness, in addition to the related oxidative stress pathway [172].

From the point of view of endothelin receptors, there seems to be some variability among the two endothelin receptors, ET_A_ and ET_B_, in vascular tone control. Although the role of ET_A_ in the vasculature appears to be clearer, the endothelial ET-1/smooth muscle ET_A_ axis plays a straightforward balancing role in maintaining the vascular tone through the release of the aforementioned vasoconstrictors, and the role of ET_B_ seems to be rather complex [131]. The ET_B_ blockade caused an increase in blood pressure; however, ET_B_ knockout mice did not develop elevated blood pressure [173,174]. Additionally, ET_B_ together with ET_A_ in the kidney has been reported to control the sodium retention function of the kidney and, in turn, cause changes in vascular tone [175]. Taken together, these results indicate that the endothelin system is essential for vascular tone control, and its dysfunction logically leads to pathological consequences.

#### 3.2.2. Clinical Implications of Endothelin in Systemic Arterial Hypertension

Clinical studies have also shown an increase in ET-1 levels in patients with systemic arterial hypertension. Elevated levels of ET-1 have been found in patients with salt-sensitive hypertension, moderate-to-severe systemic essential hypertension, hemangioendothelioma with systemic arterial hypertension, phaechromocytoma-related systemic arterial hypertension, and kidney disease, among others [2,135]. Furthermore, a single-nucleotide polymorphism in the *PHACTR1* gene, which is associated with several vascular diseases, including systemic arterial hypertension and CAD, was found to regulate the expression of ET-1 [82].

Several ERAs have been analyzed for their efficacy in the treatment of systemic arterial hypertension. The first clinical trial analyzing dual ERA bosentan in systemic arterial hypertension yielded positive results with respect to blood pressure reduction; however, several notable adverse effects occurred in the participants, notably liver enzyme elevation and fluid retention, i.e., a recurring theme for ERAs [176]. Trials in treatment darusentan, a moderately ET_A_-selective ERA, also showed improvements in blood pressure with differing levels of adverse event occurrence [177]. The relatively high level of adverse events, combined with the availability of other antihypertensive drug classes with good efficacy and fewer adverse effects, has shifted the application of endothelin-targeting drugs to resistant hypertension conditions.

Resistant hypertension, where systemic arterial hypertension persists even after three or more antihypertensive agents, including diuretics, have been administered, is a condition where ERA is currently being investigated. Several agents, especially those with high ET_A_ selectivity, are currently under clinical trial. Notable among these trials is the DORADO trial, where three different dosages of darusentan (50 mg, 100 mg, or 200 mg) could effectively reduce both seated systolic and diastolic blood pressure by at least 10 mmHg, a larger drop compared to the placebo-treated control [178,179]. However, in the DORADO-AC trial, in which an active treatment control group of guanfacine was included in addition to three different dosages of darusentan and placebo, placebo treatment unexpectedly reduced systolic blood pressure to a level similar to that of darusentan in the initial seated blood pressure measurement [180]. Only after post-hoc analysis using ambulatory blood pressure measurement did darusentan demonstrate its superiority over placebo and guanfacine, and the results of this trial put a halt in darusentan usage for treatment-resistant hypertension. A novel ERA, procitentan, a metabolite of macitentan, is currently undergoing a phase III trial (PRECISION) in resistant hypertension, buoyed by the positive efficacy results in a dose-dependent study and its overall safety profile [67,68]. The results from pre-clinical and human studies appear promising, with significant changes in blood pressure (BP) observed within 14 days. Aprocitentan enhances the effect of BP lowering by other antihypertensive drugs. In summary, procitentan exhibits protective capabilities in patients with resistant hypertension.

It is also important to note that although it is beyond the scope of this article, the kidney is another vital organ in the regulation of vascular tone, and various ERAs, especially those selectively blocking ET_A_, such as atrasentan, have been shown to be capable of treating kidney-disease-related systemic arterial hypertension. The promising results from ET_A_-selective atrasentan treatment in diabetic nephropathy patients, as shown in the SONAR trial and sparsentan (a combination of ET_A_-selective ERA and angiotensin II type 1 receptor antagonist) treatment for focal segmental glomerulosclerosis (FSGS) in the DUET trial, suggests that kidney-disease-related systemic arterial hypertension could also be an area where ERA and other endothelin-targeting treatments could be beneficial in the near future [76,181].

### 3.3. Heart Failure

#### 3.3.1. Endothelin and Heart Failure (HF)

Heart failure due to various etiological causes has long become a topic of interest in relation to endothelin, as it was later shown in various studies that the effects of endothelin do not stop merely at vasoconstriction. Indeed, due to the wide range of molecular and cellular effects that could be mediated by the binding of ET to its canonical receptor, several molecular pathways are important to cardiomyocyte hypertrophy and heart remodeling, such as phosphoinotiside 3-kinase (PI3K)/protein kinase B (AKT)/glycogen synthase kinase 3 beta (GSK3β), mitogen-activated protein kinase (MAPK) 1/2, transforming growth factor beta (TGF-β), nuclear factor kappa B (NFkB), caspases, natriuretic peptides, and protein kinase C (PKC), among other pathways [182,183,184,185,186].

Of note, as previously mentioned, several pathways and process have been heavily connected with the endothelin system, especially in relation to the development and progression of chronic HF. The renin–angiotensin–aldosterone system has long been identified as a target pathway affected by endothelin activation. Indeed, different changes in ET-1-induced RAAS activation occur during the initial development of chronic HF, where the ET-1/ET_A_ axis increased blood pressure and induced RAAS downregulation attenuated by the sympathetic nervous system, and during the progression of HF, where RAAS activation could be induced by the same ET-1/ET_A_ axis due to the cardiac output decrease [187,188]. Another pathway related to endothelin that is important in the progression of HF is the TGF-β, which has been shown to mediate the fibrotic remodeling of the cardiomyocyte [183]. Inflammatory pathways are also another vital part of endothelin-induced factor causing progression of chronic HF. Inflammatory cytokines (e.g., TNF-α, interferon-γ, IL-1β, IL-6) are overproduced in the presence of ET-1 overexpression via NFkB translocation and induction of target cytokines expression, all of which leads to inflammatory cells infiltration and dilated cardiomyopathy phenotype in mice [189]. These are just some of the examples demonstrating the wide range of influence that the endothelin system could have on chronic HF development. The role of endothelin on various etiological causes of chronic HF has been extensively studied throughout the years in various experimental models of HF. Notably, diabetic cardiomyopathy has gained significant interest as a condition in which endothelin plays a significant role. Widyantoro et al., using a streptozotocin mouse model, previously demonstrated that ET-1 is important in the endothelial-to-mesenchymal transition process, which is important in the pathology of diabetic cardiomyopathy [183]. Hypertrophic cardiomyopathy is another condition where ET-1 is known to play a role in inducing its pathological phenotypes. Induced pluripotent stem cell (iPSC)-derived cardiomyocytes isolated from hypertrophic cardiomyopathy treated with ET-1 showed marked hypertrophy and myofibrillar disarray [190]. In addition, pressure overload mouse models due to transverse aortic constriction (TAC) have also shown that ET-1, especially those originating from the vasculature, play an important role in hypertensive myocardial hypertrophy [191]. The development of pacing-induced chronic HF model in dogs is another process where the ET-1/ECE-1/ET_A_ axis has been proven to be important. Two different studies showed that chronic treatment with ET_A_ antagonist or with ECE inhibitor could ameliorate the HF phenotype [192,193]. Chronic HF due to ischemic heart disease has been well documented to relate heavily with ET-1 overexpression, such as those found in the failing hearts of rats after prolonged coronary artery ligation, while prolonged treatment with ERAs that selectively target ET_A_ was reported to improve this condition [194,195]. Changes in both the peptide and in the receptors also occur in ischemic heart-disease-related chronic HF. An increase in ET_A_ and ET_B_ receptor expression in the coronary arteries was also observed in ischemic heart disease-caused chronic HF, and in another study, the chronic blockade of these receptors could attenuate left ventricular dysfunction and dilation in rats, which, in part, became the basis of the following clinical trials of ERA on chronic HF condition [196].

In short, the link between all etiological causes of HF is that endothelin, specifically ET-1, plays an important role in heart remodeling through the modulation of inflammation, apoptosis, and fibrosis [197].

#### 3.3.2. Clinical Evidence of Endothelin in Chronic and Acute Heart Failure

As a biomarker, blood ET-1 levels, including the active and modified forms of ET-1, have prognostic value in predicting hospitalization and mortality for both heart failure with reduced and preserved ejection fractions [198,199]. Similar to other cardiovascular conditions, ET-1 levels are elevated in the blood samples of chronic HF patients of various etiologies, including diabetic cardiomyopathy and hypertrophic cardiomyopathy [200,201]. One study related natriuresis, fluid congestion, and poor clinical prognosis to the elevation of plasma ET-1 levels, while another indicated that a higher ET-1 plasma level at admission is a prognostic marker for poor short-term prognosis in acute heart failure (HF) [202,203]. Chronic HF due to ischemic heart disease has also been correlated clinically with ET-1. The correlation between ET-1 levels and inflammation has also been established in clinical setting. For instance, a study established ET-1, in addition to adrenomedullin, to be correlated to inflammation in chronic HF condition [204]. Additionally, it has also been found that ET-1 levels is elevated together with other inflammatory cytokines (TNF-α, IL-6, and MCP-1) in the macrophages of chronic HF patients [205]. Other inflammatory markers important in chronic HF, such as the C-reactive protein (CRP) and NLRP3, have also been correlated with ET-1 [206,207,208,209]. As inflammation has been established as an important pathway that modulates the pathophysiology of chronic HF, the link between endothelin and inflammation becomes essential in devising ways to target endothelin dysfunction as a therapy [208]. All of the evidences above show that ET-1 elevation is an established marker for pathological conditions such as chronic HF.

Unfortunately, clinical trials of ERAs in this condition have either not found satisfactory positive results or were halted prior to achieving the endpoint due to the high occurrence of side effects. Unlike PH and hypertension, although the importance of endothelin in the pathophysiology of various etiological causes of HF both in chronic and acute setting is undeniable, ERA in HF does not enjoy a similar level of clinical success and is currently, in essence, not favored by other drugs targeting different pathways. One of the earliest reported trial for ERA usage in chronic HF condition, The REACH-1 trial, which first analyzed the use of bosentan in chronic HF, was discontinued because of the high occurrence of elevated liver enzyme levels [210]. Similarly, the ENABLE trial that utilized a lower bosentan dosage was halted because of the fluid retention caused by bosentan, a paradoxical effect that is not beneficial for the patients [211]. Other ERAs, such as darusentan in the EARTH trial and enrasentan in the ENCOR trial, did not achieve positive results for ERA treatment in chronic HF [2,212]. Furthermore, recent results from the SONAR trial revealed similar fluid retention problems in atrasentan; however, in a post-hoc analysis, the kidney protection benefit was deemed to outweigh the fluid retention problem [213].

Acute heart failure (acute HF) is another focus of study for the clinical use of ERA. However, similar to chronic conditions, no encouraging results have emerged in this field. The RITZ-1 and RITZ-2 sets of clinical trials attempted to answer the question of whether ERA, in this case the non-selective tezosentan, could be useful in an acute HF setting. Conflicting efficacy results were obtained between RITZ-1, which found no impact of tezosentan on clinical symptoms and cardiovascular events, and RITZ-2, which showed hemodynamic and symptom improvements after tezosentan in severe chronic HF [214,215]. This conflicting result was one of the main reasons for the shelving of tezosentan as a medication for heart failure. Similarly, the VERITAS trial observed minimal clinical effects of tezosentan treatment in acute HF [216]. The discrepancy seen between the successful preclinical studies and disappointing clinical trials could be factored by various causes. These include the administration of other standard-of-care HF treatment in conjunction with ERA treatment in patients that might have overlapping beneficial effect with ERA—a phenomenon that obviously cannot be found in the animal models that only received ERA. Another possible explanation is that the differences in ERA treatment effects (and side effects) between humans and rodents or other animals used as experimental models were physiologically significant enough to elicit unwanted side effects in other untargeted organs only in humans. As such, novel modes of therapy are needed to properly address the need to alter the effects of endothelin on HF. Encouragingly, several clinical studies are being conducted to analyze the perceived “gap” in endothelin importance in HF pathophysiology and the ineffectiveness of ERA (NCT 02319590, NCT02124824), with the hope that novel strategies can be implemented to counter the dysfunction of endothelin system in HF. At the same time, treating the phenomenon that is associated with endothelin system activation, such as targeting the inflammation of interleukins, could be beneficial as an alternative.

### 3.4. Atherosclerosis, Acute Coronary Syndrome and Coronary Artery Disease

#### 3.4.1. Endothelin and Coronary Artery Pathologies

Endothelin has also been implicated in the pathophysiology of atherosclerosis and other CAD, including, but not limited to, vasospastic angina, microvascular angina, prinzmetal angina, and Takotsubo syndrome [2,217,218,219]. Specifically pertaining to atherosclerosis, the relation of the classical risk factors to develop atherosclerosis (diabetes, obesity, smoking, arterial hypertension) with an increased level of endothelin, specifically ET-1, has demonstrated the correlation between ET-1 and atherosclerosis [220]. ET-1 is also found with increased expression at various sites and cells of atheroma plaques, such as in fresh coronary thrombi of patients with ST-segment elevation myocardial infarction (STEMI) and vascular smooth muscle cells of atherosclerotic coronary arteries [217,221]. The ischemia–reperfusion injury mice model via coronary artery ligation and reperfusion also revealed an increase in plasma ET-1 levels after injury, while the blockade of ET_A_ could attenuate the myocardial injury via NO-related mechanism [222]. More recently, the ERA tezosentan was also found to be effective in attenuating ischemia–reperfusion-induced left ventricular remodeling in rats [223].

Endothelin also plays a role in immune cells and immunological processes that correlate with the formation of atherosclerotic plaques. ET-1 is known to be pro-inflammatory because of its ability to activate macrophages and release inflammatory cytokines, including TNF-α, IL-6, and IL-1β, while also increasing adhesion molecule expression and stimulating neutrophil aggregation [224]. These effects actually cause a reciprocal induction of ET-1, specifically from the pro-inflammatory cytokines, causing a vicious inflammatory cycle that promotes further vascular injury, thereby promoting atherosclerosis [217,225]. This was confirmed in an in vivo model of high-fat diet-induced atherosclerotic ApoE knockout mice, where overexpression of endothelin ET-1 led to the exacerbation of atherosclerotic lesions and, concurrently, an inflammatory phenotype [226]. This inflammatory phenotype also extends in the event of acute myocardial infarction. As inflammatory cytokines and inflammasomes, such as IL-6, IL-1β, or NLRP3, has been established to be major players in the condition of acute myocardial infarction, and ET-1, as has been mentioned previously, correlates strongly with inflammation, both as an inducer of cytokine expression and as one of the secreted factors after exposure to inflammation [208,227].

During the formation of the atherosclerotic plaque or during the infarction event, endothelin also plays a role in the post-infarction process of left ventricular repair and remodeling. In relation to the previously mentioned chronic HF due to myocardial injury, there is also evidence of endothelin involvement in the acute post-infarction phase of myocardial remodeling. Specifically, it is implied in the EPHESUS study that an elevation of bigET-1 could be found in the post-infarcted heart [228]. In the animal model, it has already been known that ET-1 levels in the early days of post-infarction correlate with left ventricular remodeling. Interestingly, while the blockade of ET receptors could prove beneficial to attenuate left ventricular remodeling and improve its function post-infarction, as previously mentioned, another study showed that the very early blockade of the same receptors could paradoxically aggravates left ventricular remodeling, implying the importance of endothelin system activation in the acute post-infarction response of the heart [196,229]. Molecularly, some of the molecular pathways previously mentioned in other sections of this review, such as NO, RAAS, and inflammatory pathways, among others, are both important and closely related to endothelin system activity, particularly to ET-1 [230,231,232]. As such, it is clear that endothelin has also been extensively studied in the field of CAD and it is an important pathway to tackle this condition.

#### 3.4.2. Clinical Application of Endothelin in Coronary Artery Disease

In the clinical setting, many studies have found a correlation between ET-1 and CAD and atherosclerosis [221]. A recent study found that high ET-1 levels are increased in atherosclerotic arteries and could reflect the severity of three-vessel disease [233]. Similarly, another study found that big ET-1 levels were increased in patients with CAD [234]. Furthermore, in patients undergoing CABG, ET-1 is known to be elevated in patients with diabetes compared with non-diabetic patients [235]. Meanwhile, a clinical trial evaluating eplerenone (EPHESUS), which is known to have anti-inflammatory capability, found that blood levels of bigET-1 could also be reduced by eplerenone treatment [228]. The role of ET-1 in left ventricular remodeling post-infarction is also confirmed by the elevation of its plasma level in the acute post-myocardial infarction phase of patients with acute myocardial infarction [236]. In addition to the peptide, its enzyme, ECE-1, is also found to increase in various cells in atherosclerotic plaques, including endothelial cells, smooth muscle cells, macrophages, and the fibrous cap of the plaque, while this increase in ECE-1 is thought to be functionally relevant [237,238]. Interestingly, there are different effects of the dual ET receptor blockade and the ET_A_-specific blockade in the peripheral and coronary arteries. In contrast, in the peripheral arteries (in the case of the study conducted by Rafnsson et al.), forearm vasodilation could be best achieved by the dual ET receptor blockade; in coronary arteries, the ET_A_-specific blockade interestingly showed the most effective vasodilation [239,240]. This can, in part, explain the varying degrees of success ERA has in CAD.

Clinical trials involving ERAs in CAD include ENDORA, whereby ambrisentan treatment in NSTEMI/ACS could reduce neutrophil overactivation and hs-troponin-T levels [241]. In contrast, no effects of ERA were found in acute coronary syndrome accompanying HF, as shown by the RITZ series of trials mentioned above. In the RITZ-4 trial, the investigators focused on the use of tezosentan in the case of acute coronary syndrome-related acute heart failure [242]. Unfortunately, no apparent benefit could be found after tezosentan in comparison with placebo, and this trial dims the hope of tezosentan usage in acute coronary syndrome. Taken together with the results of the studies of ERA in HF, although it is unfortunate that the potential of endothelin as a treatable pathway has not been fully realized, it is encouraging to observe that there are specific groups of patients that could benefit from ERA treatment. Furthermore, recent advances in anti-inflammatory therapy, such as the interleukin-targeting canakinumab and anakinra, gives hope that endothelin system dysfunction could be treated through the alleviation of inflammatory phenotype [208]. Even so, further studies are still warranted to analyze the specific populations that benefit from ERA treatment in addition to devising new strategies to combat the dysfunction of this pathway.

### 3.5. Others

#### 3.5.1. Cardiac Arrythmia

Although less evidence can be seen in comparison with other conditions, endothelin also appears to play a role in several arrhythmias. The ET-1/ET_A_ axis has been shown to possess arrhythmogenic potential in various studies of cardiomyocytes through several proposed mechanisms, such as its ability to handle intracellular Ca^2+^ and MMP9-derived pro-fibrotic activity [243,244]. Atrial fibrillation, for instance, is a condition in which ET-1 and its precursor big ET-1 were found to be elevated, while the arrhythmogenic activity of the pulmonary veins could also be controlled by ET-1 [245,246,247]. In the case of ventricular arrythmia, ET-1 gene polymorphism has been identified as a risk factor in having a worse hemodynamic outcome during a ventricular arrythmia episode [248]. On the contrary, a study in isolated rat cardiomyocytes could not prove that ET-1 has a direct role in causing the arrhythmogenic properties in the ventricle [249]. In short, arrhythmia is a pathological condition where endothelin could be intervened and studied in the future.

#### 3.5.2. Antiangiogenic Treatment Adverse Effects

Antiangiogenic drugs, such as vascular endothelial growth factor (VEGF) inhibitors, have been increasingly used as treatment options to fight several forms of cancer in recent times [250]. One of its adverse events is VEGF inhibition-related hypertension [34,250]. The endothelin system has been previously related to the VEGF pathway, in which the blockade of ET receptors could improve ischemia through the VEGF-NO pathway [251]. Clinically, an increase in ET-1 plasma levels was observed after VEGF inhibition [252]. The ENDEAVOUR trial (NCT 03557190) that analyzed the use of ERA after treatment with VEGF inhibitor has been completed; however, to the best of the authors’ knowledge, no results have been published yet. We hope that endothelin could be a viable alternative to treat this specific inducer of hypertension.

#### 3.5.3. Peripheral Artery Disease

Endothelin has also been found to be involved in the peripheral artery disease development. Of note, in patients with peripheral artery disease, the blood levels of ET-1 are found to increase [253]. In the clinical setting recently, a clinical trial (the CLAU trial) indicated the possible effectiveness of ERA in treating peripheral artery disease [254,255]. In this case, bosentan was administered to patients with intermittent claudication for 12 weeks, and both the initial and four-year follow-up results indicated that bosentan treatment could be effective in improving the claudication distance, C-reactive protein (CRP) levels, and flow-mediated arterial dilation in a select group of patients with low-to-mild stages of PAD with low risk of severe adverse effects [254,255].

## 4. Future Perspectives

Dysregulation of the endothelin pathway has been identified as a cause of various diseases. Targeting ET_A_/ET_B_ receptors or their effectors has emerged as the long-term goal of developing new therapies. Over the last few decades, the development of molecular orthosteric and allosteric ligands has been the central focus of endothelin research. Notably, several emerging novel modalities targeting these receptors have been identified, such as pepducins, aptamers, and antibodies.

### 4.1. Allosteric Modulators

Allosteric modulators are molecules that can alter the biological activity of receptors through distinct binding sites of endogenous ligands. Currently, ET receptor antagonist-related adverse effects have been reported, including the risk of embryonic–fetal toxicity due to the blocking action of ET1. Allosteric modulation that reduces, but does not block, the action of ET1 may offer advantages in this regard [256,257]. The first allosteric modulator was identified in 2000, but there were no allosteric modulators that underwent a clinical trial phase until recently [258,259].

### 4.2. Peptide-Based Biased ET Receptor Signaling

Recently, in the field of G-protein-coupled receptor (GPCR) research, targeting specific downstream pathways, such as G protein or β-arrestin, via biased orthosteric ligands and/or allosteric modulators, holds a novel paradigm for targeted drug development, as depicted in Figure 3 [260]. This concept has been explored in the cardiovascular field for angiotensin II type 1 receptor (AT1R). The downstream G-protein signaling of AT1R is considered cardio-deleterious, whereas β-arrestin has cardioprotective properties. Targeted novel agonists of AT1R β-arrestin, such as TRV027, have been investigated for heart failure treatment [261,262].

Unfortunately, molecules related to ET-biased signaling have not been explored previously. However, the distinct downstream signaling properties of ET receptors offer a potential explanation for ineffective ET antagonists in cancer treatment, despite numerous studies proving that the endothelin system axis plays a significant role in cancer pathogenesis [263].

### 4.3. Pepducins/Cell-Penetrating Peptides

Pepducins are synthetic, short, cell-penetrating peptides derived from the three intracellular loops or the C-terminal tail of GPCR and ET_A_/ET_B_ receptors. The N-terminus of pepducins is lipidated to support the transfer process between the cell membrane and anchor the peptide. Once inside the cell, pepducins stabilize receptor conformations, which may stimulate or inhibit intracellular signaling [46,264]. Previous studies have revealed that these pepducins can modulate ET1 signaling capabilities and ameliorate hypoxic-induced pulmonary hypertension in rats [265,266].

### 4.4. Antibody against ET Receptors

Therapeutic vaccines are novel modalities used for the treatment of chronic diseases, including cardiovascular diseases. Compared with small-molecule drugs, vaccines and antibodies have several advantages. First, the ability of therapeutic antibodies to target antigens is highly specific, resulting in higher efficacy and reduced side effects. Second, the serum half-life of antibodies is relatively high, affecting the frequency of administration and improving patient compliance [267].

Preclinical studies of vaccine-targeting ET_A_ receptors, getagozumab, have shown promising results for the treatment of hypoxia-induced and monocrotaline (MCT)-induced pulmonary arterial hypertension (PAH) in monkey models. Currently, getagozumab is in phase 1b clinical trials [268]. In addition, a monoclonal antibody against ET_A_ receptors (ETRQβ-002 vaccine/mAb) has been developed to effectively ameliorate pulmonary arterial hypertension (PAH) in MCT-treated and SUGEN–hypoxia-induced animal models, with satisfactory safety properties [269,270]. Similar to ET_A_ receptors, antibodies targeting ET_B_ receptors, such as Rendomab-B1 and Rendomab-B4, are also available for cancer treatment, particularly for melanoma [46].

### 4.5. ET-1 Traps

Endothelin-1 traps or ET traps are molecular constructs composed of molecules that potently bind to ET-1 fused to the Fc portion of human immunoglobulin (Ig)-G1. ET traps have shown potential therapeutic effects in in vitro and diabetic animal models. ET trap administration can have beneficial effects on diabetic target organs, such as the heart and kidney. In addition, ET traps were not immunogenic and did not exhibit any adverse effects. Therefore, the ET trap is an attractive target for further therapeutic development of disease-associated pathological ET-1 [271,272].

## 5. Conclusions

Endothelin is a vital peptide with three isoforms and was originally identified as a potent vasoconstrictor. In subsequent years, the wide array of influences that the endothelin system can affect has led to evidence that demonstrates the importance of endothelin in various cardiovascular diseases, including hypertension, PH, HF, and CAD, among others. Consequently, strategies have been, and are currently being, developed to improve the ways that clinicians can target this pathway. Although significant progress has been made in the 35 years since the discovery of endothelin, many questions remain in this field, and further studies are encouraged to fully realize the potential of targeting endothelin in cardiovascular diseases.

## Figures and Tables

**Figure 1 biology-11-00759-f001:**
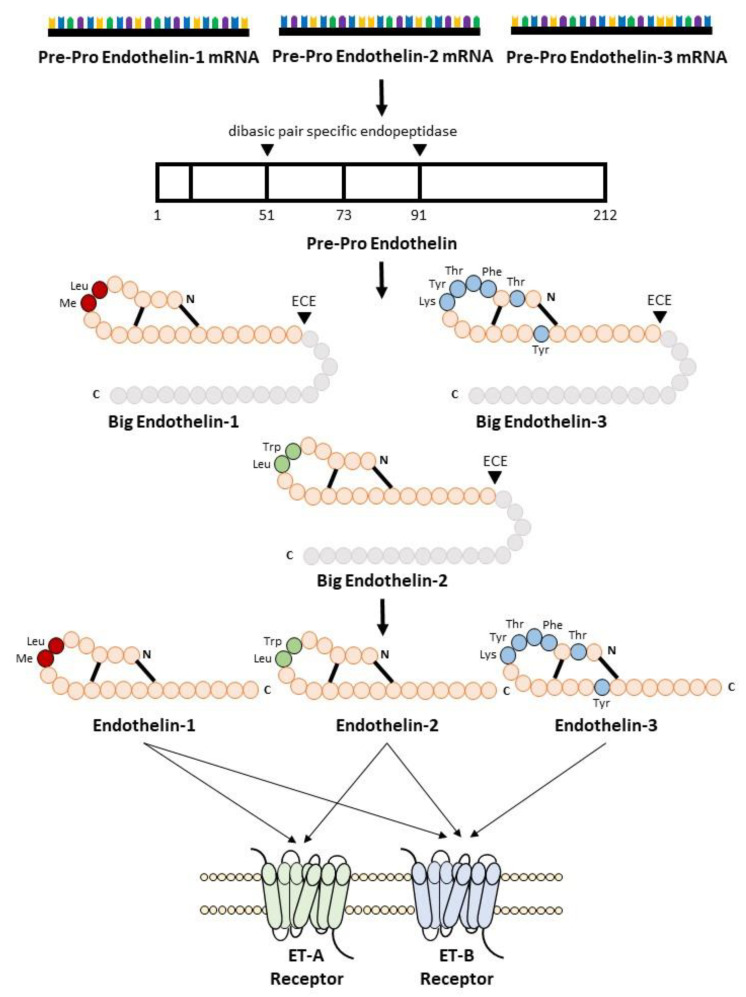
Biosynthesis of endothelin.

**Figure 2 biology-11-00759-f002:**
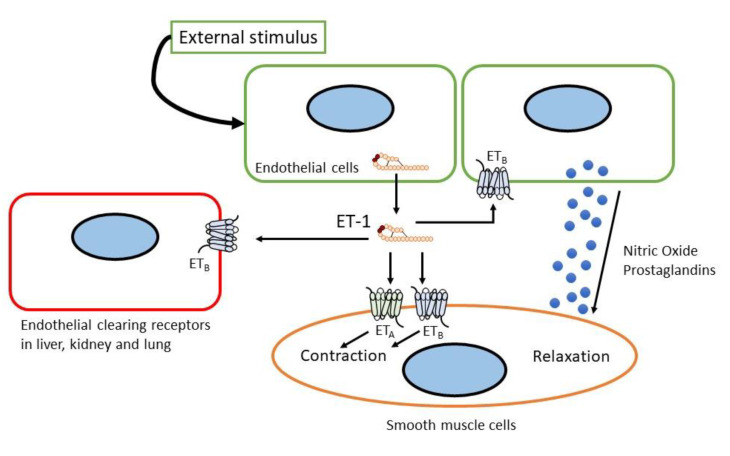
Sites and mechanism of action of endothelin.

**Figure 3 biology-11-00759-f003:**
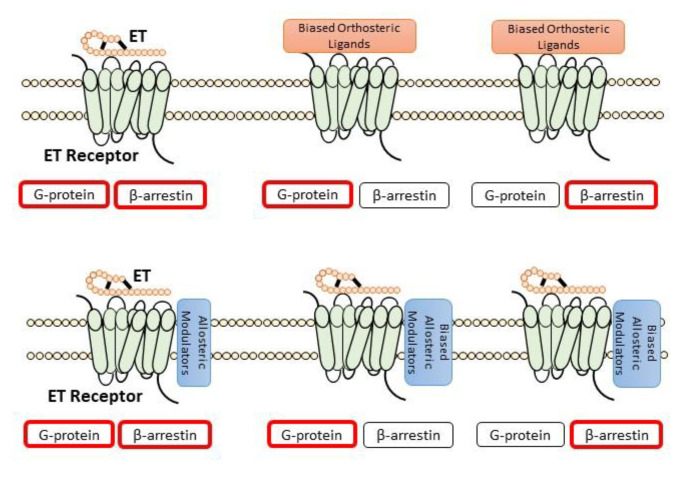
Pharmacologic mechanism of biased G-protein-coupled receptor signaling.

## Data Availability

Not applicable.

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
