# Peer review of "Endothelin and the Cardiovascular System: The Long Journey and Where We Are Going"

_biology, 2022, doi:10.3390/biology11050759_

Round 1

Reviewer 1 Report

Haryono et al. provide the overview of the role of endothelin in the pathophysiology of cardiovascular disease and treatment approaches targeting the endothelin-related pathways, as well as future perspectives for endothelin-related research in cardiovascular disease. This manuscript addresses an interesting topic. Both the rationale and scientific content of the manuscript are valuable. Overall, the manuscript is well written and easy to follow. However, I have a few concerns and suggestions to consider for improving the manuscript.

The authors summarized pretty well the existing results of experimental research and clinical studies on the role of endothelin system in the pathophysiology of PAH and systemic arterial hypertension, as well as clinical evidence on the effects of endothelin antagonists for treating these conditions. However, sections about the role of endothelin system in heart failure (HF) (especially chronic HF) and coronary artery disease (CAD) need improvements. Specifically, endothelin-related mechanisms of development and progression of chronic HF, especially related to HF of ischemic etiology, require more attention. Also, endothelin-related aspects of a development of left ventricular dysfunction and remodeling in the course of CAD, e.g., post myocardial infarction, need attention and should be included in the manuscript, especially given the title (i.e., „Endothelin and the Heart”). In addition, interactions between activation of endothelin system and inflammatory pathways in post-infarct myocardial injury/dysfunction/remodeling and chronic HF, all of which have been commonly observed in previous studies, require further attention and clarification, especially given recent advancements in anti-endothelin and anti-inflammatory treatments. I suggest adding at least a paragraph devoted to these topics and including relevant references. Example relevant references that would fit in this context are:

Olivier, A., et al. Int J Cardiol. 2017; 241: 344-350.

Gombos T., et al. Inflamm Res. 2009 Jun;58(6):298-305.

Abbate, A., Circ. Res. 2020, 126, 1260–1280.

Toldo, S., Abbate, A. Nat. Rev. Cardiol. 2018, 15, 203–214.

Swiatkiewicz, I. et al. Int. J. Mol. Sci. 2020, 21, 807.

ÅšwiÄ…tkiewicz, I., et al. Int. J. Mol. Sci. 2021, 22, 3169.

Gottlieb, S. S., et al. Clin. Biochem. 2015, 48(4-5), 292–296.

Sikkeland, L.I., et al. PLoS One. 2012;7(5):e 36815.

Everett, B.M., Ridker, P.M, et al. Circulation 2019, 139, 1289–1299.

Ridker, P.M., et al. Eur. Heart J. 2020, 41, 2153–2163.

Also, I would suggest using “systemic arterial hypertension” or “systemic essential hypertension” rather than “hypertension” in sections on arterial hypertension. In the section about HF, I suggest including also the term “chronic HF”. Peripheral artery disease is not a type of CAD, so should not be included in the section on CAD.

The manuscript requires editing corrections. For example, the numbering of subsections titles is erroneous. Also, the use of abbreviations and associated full names should be consistent throughout the manuscript rather than using them interchangeably in seemingly random fashion (e.g., acute heart failure or AHF, coronary artery disease or CAD, etc.). Also, all abbreviations should be explained when used for the first time in the text (e.g., STEMI, ACS, NSTEMI, etc.).

Reviewer 2 Report

This review is thorough and well written. The comments below are intended to improve the overall quality of the product.

Title: Rather than using heart in the title, cardiovascular system would be more accurate considering the review highlights both cardiac and vascular effects of endothelin

Since hypertension was discussed, it would be more comprehensive to also discuss endothelial function and arterial stiffness

Figure 2 – text should me larger. A majority of the text is currently unclear

Additional figures would be helpful in order to break up the text. Furthermore, a visual representation of future directions would also be helpful

Last sentence of the abstract – should be “the future”

Round 2

Reviewer 1 Report

The authors adequately addressed my comments.

Reviewer 2 Report

Thank you for taking the time to thoroughly address previous comments. I believe the manuscript has been significantly improved.
